# The Role of EUS and EUS-FNA in Differentiating Benign and Malignant Gallbladder Lesions

**DOI:** 10.3390/diagnostics11091586

**Published:** 2021-08-31

**Authors:** Susumu Hijioka, Yoshikuni Nagashio, Akihiro Ohba, Yuta Maruki, Takuji Okusaka

**Affiliations:** Department of Hepatobiliary and Pancreatic Oncology, National Cancer Center Hospital, Tokyo 104-0045, Japan; yonagash@ncc.go.jp (Y.N.); aohba@ncc.go.jp (A.O.); ymaruki@ncc.go.jp (Y.M.); tokusaka@ncc.go.jp (T.O.)

**Keywords:** Endoscopic ultrasonography (EUS), EUS fine-needle aspiration (EUS-FNA), gallbladder carcinoma, xanthogranulomatous, cholecystitis gallbladder lesions

## Abstract

Endoscopic ultrasonography (EUS) has greater spatial resolution than other diagnostic imaging modalities. In addition, if gallbladder lesions are found and gallbladder cancer is suspected, EUS is an indispensable modality, enabling detailed tests for invasion depth evaluation using the Doppler mode and ultrasound agents. Furthermore, for gallbladder lesions, EUS fine-needle aspiration (EUS-FNA) can be used to differentiate benign and malignant forms of conditions, such as xanthogranulomatous cholecystitis, and collect evidence before chemotherapy. EUS-FNA is also useful for highly precise and specific diagnoses. However, the prevention of bile leakage, an accidental symptom, is highly important. Advancements in next-generation sequencing (NGS) technologies facilitate the application of multiple parallel sequencing to EUS-FNA samples. Several biomarkers are expected to stratify treatment for gallbladder cancer; however, NGS can unveil potential predictive genomic biomarkers for the treatment response. It is believed that NGS may be feasible with samples obtained using EUS-FNA, further increasing the demand for EUS-FNA.

## 1. Introduction

Endoscopic ultrasonography (EUS) plays a major role in the diagnosis of gallbladder lesions. The digitization of diagnostic ultrasound imaging devices and advancements in ultrasound contrast media help obtain the blood flow information with the Doppler-mode and contrast-enhanced EUS, and more detailed EUS investigations can be performed to determine the presence of gallbladder lesions and invasion depth for suspected gallbladder cancer. In some cases, EUS combined with fine-needle aspiration (EUS-FNA) is also useful for distinguishing between benign and malignant gallbladder lesions. This review outlines the use of EUS in the diagnosis of gallbladder lesions.

## 2. EUS for Gallbladder Lesions

Table 1 presents the classification of gallbladder lesions. Gallbladder lesions are classified as neoplastic or non-neoplastic based on their microscopic structure and invasive characteristics (Table 1). In cases of suspected gallbladder cancer, clinical practice guidelines for the management of biliary tract cancers [1] recommend diagnostic imaging, including EUS, as the third diagnostic step for all gallbladder cancers.

EUS is performed using an endoscope equipped with an ultrasound probe. High ultrasound frequencies (5 mHz to 20 mHz) are used for EUS, and it has a high spatial resolution, thereby facilitating a detailed examination of the gallbladder because it can approach and examine the organ at a closer range than the conventional US [2,3,4]. This helps in the qualitative diagnosis of lesions and evaluation of tumor invasion depth [5]. However, because the position of the gallbladder can differ between individuals, visualizing the entire gallbladder is occasionally difficult, particularly the gallbladder fundus. For this reason, different imaging modalities, such as abdominal ultrasonography (US), computed tomography (CT), or magnetic resonance cholangiopancreatography (MRCP), must be used before EUS is performed.

There are two types of EUS scopes for radial scanning and convex array. The devices used for these provide different types of images and are therefore used in various visualization methods. Kaneko et al. [6] reported that there was no significant difference in visualization between these two types of devices in the examination of the pancreaticobiliary region.

## 3. Diagnosis of Gallbladder Tumor

### 3.1. Diseases That Require Differentiation from Early Gallbladder Cancer

A pedunculated type of gallbladder tumor is visualized in EUS as a polyp-shaped or papillary elevated lesion with a sessile (broad-based) or peduncle smooth surface and a uniform internal echo pattern (Figure 1). Polyp-shaped lesions of gallbladder cancers are normally early-stage cancers with a wall invasion depth reaching the mucosal layers [7]. Gallbladder cancers sometimes sequentially spread extensively around the main lesion. Tumor diameter is also an important aspect, and a study of gallbladder polyps >10 mm in diameter reported that a single polyp (OR, 3.680–3.856) and polyps of a larger size (OR, 1.450–1.477) were independently associated with neoplastic polyps (*p* < 0.05). In single polyps or polyps >14 mm, the sensitivity for differentiating neoplastic from non-neoplastic polyps was 92.3% [8]. This theory was based on the evidence of an observational study that showed some adenomas having malignant components [9], along with the adenoma–carcinoma sequence of colorectal polyps [10,11]. The Joint European guidelines recommend cholecystectomy for polyps ≥10 mm [12]; however, 5% of lesions ≤10 mm are also known to be malignant [13].

Diseases that require a differential diagnosis from pedunculated-type gallbladder cancer include cholesterol polyps, gallbladder adenomas, inflammatory polyps, and fibrous polyps (Table 1). Cholesterol polyps account for approximately 95% of raised gallbladder lesions and occur when histiocytes (foamy cells) ingest cholesterol esters, accumulate under the mucosal epithelium, and expand to form polyp-shaped growths. These polyps can break off, leading to complications similar to those caused by small gallstones [14]. During EUS, cholesterol polyps typically appear as multiple lesions and are homogeneous, pedunculated, and smaller than ≤4 mm; polyps are also generally more hyperechoic than the liver parenchyma or gallbladder wall [12,14,15]. However, as the cholesterol polyp grows, the echo intensity inside the polyp may decrease; when this occurs, the cholesterol polyp cannot be easily differentiated from adenoma or cancer. Another study on gallbladder polyps ≥10 mm used contrast-enhanced EUS to diagnose benign or malignant polyps based on the presence of an irregular contrast pattern and reported a 90.3% sensitivity and 96.6% specificity for this method [16].

Gallbladder adenoma, which can exhibit premalignant behavior (Figure 2), is a rare condition that accounts for 10% of ultrasonographically diagnosed gallbladder polyps [14]. It is usually solitary, 5–20 mm in size, and can be sessile or pedunculated [14,17,18]. Gallbladder adenomas have four histological types: pyloric, intestinal, foveolar, and biliary. Histologically, 70% of gallbladder adenomas are tubular adenomas of the pyloric gland. Pyloric gland tubular adenoma of the gallbladder is characterized by multiple microcysts inside a polyp; however, it is difficult to distinguish between cancer and this type of adenoma.

Gallbladder adenomas are generally homogeneous polyps, often isoechoic, contain liver parenchyma, and are sessile or pedunculated. An intralesional vascular spot may be observed on color Doppler investigation [14]. Using contrast-enhanced EUS, pyloric gland tubular adenoma of the gallbladder tends to exhibit more uniform contrast enhancement than gallbladder cancer or cholesterol polyps [19]. Gallbladder adenomas are prone to cancer development by the adenoma–carcinoma sequence; however, the frequency of adenomas progressing to adenocarcinomas remains unclear [12,14,15,18,20]. Intracystic papillary neoplasm (ICPN) is a gallbladder lesion concept proposed by the 2010 WHO classification (Figure 3), although some aspects of how an ICPN differs from adenoma are unclear, and a unified pathological opinion on the topic is anticipated.

Pancreaticobiliary maljunction (PBM) is a risk factor for biliary tract and gallbladder cancers as it causes mixing of the pancreatic fluid and bile. In particular, gallbladder cancer complicates 40% of PBM cases not associated with dilatation of the extrahepatic biliary tract (undilated PBM). Hyperplastic changes in the gallbladder epithelium are found in 38–63% of PBM cases [21], and biliary tract dilatation is not observed in a high percentage of PMB cases (91–100%) [22]. During the EUS procedure, PBM must always be ruled out when diffuse thickening of the gallbladder wall due to hyperplastic changes is observed by EUS (Figure 4).

### 3.2. Diseases That Require Differentiation from Advanced Gallbladder Cancer

Adenomyomatosis (ADM) and cholecystitis typically require differentiation from advanced gallbladder cancer. ADM can be diagnosed using EUS as wall thickening with a uniform surface, and, inside the thickened wall, as microcystic anechoic areas that indicate the presence of Rokitansky–Aschoff sinuses (RAS) and the “comet tail” artifact, which is a form of reverberation. Cancer can also coexist with ADM, and EUS must be carefully performed to identify the presence of irregular unevenness on the ADM surface.

Chronic cholecystitis and xanthogranulomatous cholecystitis (XGC) manifest as a wide variety of imaging findings. XGC is a subtype of chronic cholecystitis in which granulomas form due to histiocytes in the gallbladder wall ingesting purulent bile leaked from bile-filled RAS after the intra-gallbladder pressure increases due to various causes (i.e., incarcerated gallbladder neck stone and cancer). XGC is an intramural infection: it causes marked gallbladder wall thickening, spreads inflammation to surrounding organs, and manifests with findings that resemble advanced gallbladder cancer (Figure 5). Typical imaging-based findings of XGC are a linear enhancement on contrast-enhanced CT, indicating an intact gallbladder mucosal surface, and intramural high signal intensity on T2-weighted MRI, indicating intramural abscess. Although a definite diagnosis of XGC is difficult [23], EUS-FNA is the preferred modality when necessary.

## 4. Staging of Gallbladder Cancer

An important role of EUS in gallbladder cancer diagnosis is to determine the invasion depth. EUS is a useful tool for assessing the tumor depth of invasion [24,25].

Fundamental to the assessment of invasion depth is interpreting the layer structure of the gallbladder wall visualized by EUS. The gallbladder wall is divided into the mucosa (M), muscularis propria (MP), subserosa (SS), and serosa (S) layers (Figure 6). In EUS, the gallbladder wall is visualized as three layers: the hyperechoic, hypoechoic, and hyperechoic layers from the lumen side [26]. The first hyperechoic layer is a boundary echo. The second hypoechoic layer includes M, MP, and SS fibrous layers (shallow layer), and the hyperechoic outer layer includes the SS fatty layer (deep layer) and serosa. It should be noted that early gallbladder cancer (T1) cannot be diagnosed if the outer hyperechoic layer is identified as normal on EUS images (Figure 7), because the hypoechoic layer includes the SS fibrous layer (shallow layer). Therefore, T1 and T2 lesions cannot be easily differentiated using EUS. Conversely, “r” disappearance or thinning sign on the hyperechoic outer layer suggests invasion beyond the deep SS layer (Figure 8).

In recent years, attempts have been made to improve T-staging and invasion depth accuracy by performing contrast-enhanced EUS (Sonazoid^®^ or SonoVue@) and EUS elastography to assess the depth of fibrosis, respectively [27].

## 5. EUS-FNA of the Gallbladder

Obtaining tissue samples from gallbladder lesions before surgery is difficult, and diagnostic imaging is the main method for differentiating between benign and malignant gallbladder lesions. Although the diagnostic systems for gallbladder lesions are improving owing to advancements in diagnostic imaging technology, differentiating between benign or malignant lesions remains difficult in a significant number of cases, and tissue diagnosis is necessary to rule out other treatable conditions, including not only benign disease but also lymphoma and tuberculosis.

Bile cytodiagnosis using transpapillary endoscopic gallbladder drainage (ENGBD) is reportedly a useful endoscopic method for the cytological diagnosis of gallbladder lesions. However, there are some issues regarding ENGBD cytodiagnosis. This technique poses the risk of perforating the gallbladder duct with the guidewire. In contrast, if a gallbladder lesion can be visualized as a tumor by EUS, a pathological diagnosis can be established using EUS-FNA [28].

Despite the widespread use of EUS-FNA in various pancreaticobiliary lesions, published data regarding its role in gallbladder mass lesions are scarce [23,29,30,31,32]. Table 2 highlights the diagnostic performance of EUS-FNA for gallbladder lesions in previous studies. Studies on EUS-FNA for gallbladder lesions report a sensitivity of 80–100%, specificity of 100%, and accuracy of 83–100% [23,29,30,31,32]. One meta-analysis of nine studies concluded that EUS-FNA is an accurate and safe method for the evaluation of gallbladder masses, with a combined sensitivity of 84% [33]. None of the studies reported complications in the form of bile leak, cholangitis, or bleeding, suggesting that EUS-FNA is a safe procedure. This may be due to close contact of the gallbladder with the duodenum, with there being no need to go through the gallbladder lumen during the procedure [29]. However, we have to pay attention to the high sensitivity and specificity rates because of the possibility of a selective outcome reporting bias.

### 5.1. EUS-FNA Indications for Differential Diagnosis of Gallbladder Lesions

The risk of bile leakage must be carefully considered when performing EUS-FNA for gallbladder lesions. Bile leakage may not only cause peritonitis but also peritoneal dissemination in cases of malignancy. Recently, cases of dissemination needle-tract seeding caused by EUS-FNA have been reported, most of which are needle-tract seeding being intragastric wall metastases, and peritoneal dissemination is rare [35,36,37]. Though the frequency of dissemination is extremely low, peritoneal dissemination via EUS-FNA is an issue because it impairs patient survival [37]. Thus, EUS-FNA is only indicated for gallbladder lesions when the lesion can be aspirated without passing the needle through the gallbladder lumen or lesions with wall thickening. Accordingly, EUS-FNA is not indicated for gallbladder polyps or other lesions that project on the luminal side of the gallbladder. However, a recent case report described a case of gallbladder polypoid lesions successfully treated with EUS-FNA using 22-gauge needles with no complications [38].

Recently, a through-the-needle microforceps device was developed for EUS-guided tissue sampling of pancreas cystic lesions [39]. These microforceps can be advanced through the lumen of a standard 19-gauge EUS-FNA needle for through-the-needle tissue biopsy (TTNB) of pancreas cystic lesions [40,41]. EUS-TTNB is a useful tool for the differential diagnosis of pancreatic cystic lesions. It can also be applied to lesions in the gallbladder, although there is no report so far. However, about the complications, one technical review noted complications ranging from 2 to 23% [41,42]. Moreover, EUS-TTNB is thought to have the risk of gallbladder perforation because the gallbladder itself is cystic, and the gallbladder wall is thin. Therefore, we think that the indication of EUS-TTNB for the gallbladder is very restrictive at present.

Furthermore, in Japan, unlike the diagnosis for pancreatic cancer, preoperative diagnosis by EUS-FNA is not routinely recommended for resectable gallbladder cancer, because there is no evidence of neoadjuvant chemotherapy. Nevertheless, EUS-FNA should be considered in some cases of resectable gallbladder cancer, for example, when it is difficult to categorize a lesion as benign or malignant, or when the surgery is extremely invasive [23]. It can be especially difficult to differentiate XGC from gallbladder cancer, and resection with extensive hepatectomy is sometimes performed for such cases. Preoperative pathological evidence is more important for determining treatment strategies in older patients and those who pose a high risk for surgery. Nevertheless, the absence of malignant cells on EUS-FNA cannot completely rule out coexisting gallbladder cancer, and interpretation of negative EUS-FNA results for malignancy must be performed carefully. XGC coexists in 2–15% of gallbladder cancer cases [43,44]. Cases of coexisting XGC and gallbladder cancer can also involve gallbladder neck cancer with XGC in the body and fundus of the gallbladder due to elevated intra-gallbladder pressure and RAS rupture. For this reason, the gallbladder neck and cystic duct must be examined carefully in addition to the lesion area. EUS allows for the detailed examination of both the gallbladder neck and duct and reduces the risk of overlooked coexisting gallbladder cancer in cases of XGC. Surrounding lymph nodes are also enlarged in 80% of cases of advanced gallbladder cancer [34]. Thus, if enlarged lymph nodes are found when attempting to differentiate between XGC and gallbladder cancer, false positives can be reduced by performing EUS-FNA on the lymph nodes.

### 5.2. EUS-FNA and EUS-FNB for Evidence of Advanced Gallbladder Cancer

EUS-FNA is actively performed in cases of unresectable gallbladder cancer [34]. It is necessary to perform EUS-FNA to help select a treatment strategy in cases of unresectable gallbladder cancer, because small-cell carcinoma of the gallbladder (neuroendocrine type) comprises 2.5% of gallbladder cancer cases [34], and although rare, some reports have described metastatic sarcoma and malignant lymphoma of the gallbladder. Tissue diagnosis is necessary for potentially untreatable gallbladder cancer and to rule out other treatable conditions such as lymphoma, tuberculosis, and XGC [29].

Advanced gallbladder cancer is generally associated with liver metastasis, lymph node metastasis, and invasion of the surrounding tissue; EUS-FNA is a relatively easy method of obtaining histological evidence prior to chemotherapy from the primary site, liver metastases, and enlarged lymph nodes. In cases where lesions have invaded the liver, it is recommended to use either the liver parenchyma or the gallbladder wall in contact with the liver parenchyma as the invasion site [5].

Recently, novel fine-needle biopsy (FNB) devices have been specifically designed to obtain histological specimens [4,45]. Histological specimens with preserved architecture are easier to be interpreted by a standard pathologist than cytological smears [46].

Indeed, recent results of EUS-FNB for both pancreatic and non-pancreatic lesions demonstrated that the histological yield and accuracy rate are excellent [46,47,48,49,50]. Therefore, it is recommended to use EUS-FNB, in particular, for the examination of cases that require IHC, including the metastasis from another primary organ’s malignancy/a malignant lymphoma, and those whose genetic analyses have been described below.

### 5.3. EUS-FNA for Gallbladder Lesions in Clinical Practice

When taking a sample from the gallbladder, it is important that the needle does not pass through the lumen of the gallbladder because bile leakage from the needle puncture site may cause peritonitis and peritoneal dissemination in cases of malignancy. If possible, it is easier to approach the puncture site from the duodenal bulb. The gallbladder is highly mobile; hence, a needle stroke can often be difficult even when a puncture is possible. Thus, it is best to puncture the neck side of the gallbladder or gallbladder wall in contact with the liver parenchyma, in cases where this is possible. Recently, Tamura et al. reported the utility of EUS-FNA with contrast-enhanced harmonic imaging (EUSFNA-CHI), which allows the appropriate positioning of the needle within the gallbladder tumor by avoiding the fluid space, for obtaining a higher volume of tissue [51].

## 6. Future Prospects for EUS-FNB for Gallbladder Lesions

As research moves toward the realization of truly individualized medicine, the increasing popularity of next-generation sequencing will enable drug selection based on genome biomarkers. The genetic analyses of EUS-FNB specimens from some organs using targeted amplicon sequencing have already been reported. Driver genes, such as the *ERBB2*, *PIK3CA*, *IDH1/2*, *BRCA1/2,* and *FGFR2* fusion genes, have been identified in gallbladder and biliary tract cancers [52,53]. Specimens obtained by EUS-FNB can be used for next-generation sequencing of bile duct cancer. With next-generation sequencers, EUS-FNB should also become increasingly essential in the diagnosis and treatment selection for gallbladder cancers [54,55,56].

## Figures and Tables

**Figure 1 diagnostics-11-01586-f001:**
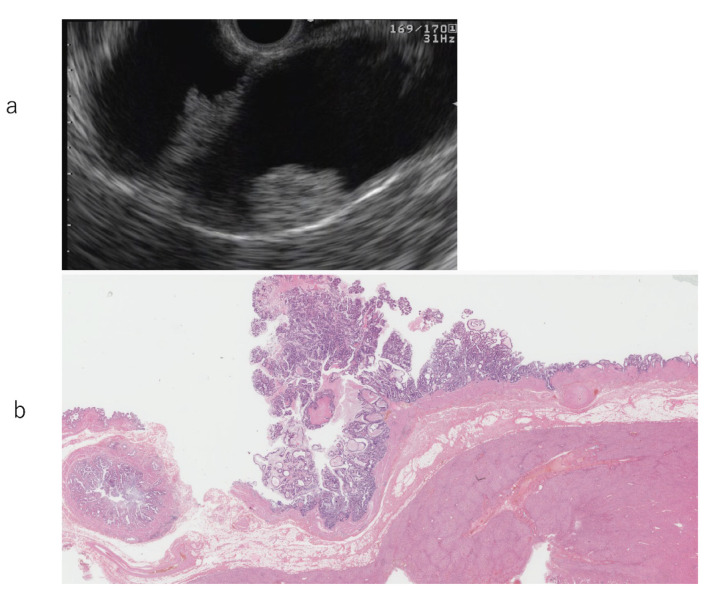
Early-stage gallbladder cancer (m) type I. (**a**) Endoscopic ultrasonography: Broad-based elevated lesion in the gallbladder body. Nodular surface structure and slightly non-uniform internal echo. Regular hyperechoic external layer of the gallbladder wall where the lesion is attached. (**b**) Magnified image: Papillary-shaped tumor. Low papillary-shaped lesion contiguous with the base of the papillary-shaped lesion. No invasion of the proper muscular layer.

**Figure 2 diagnostics-11-01586-f002:**
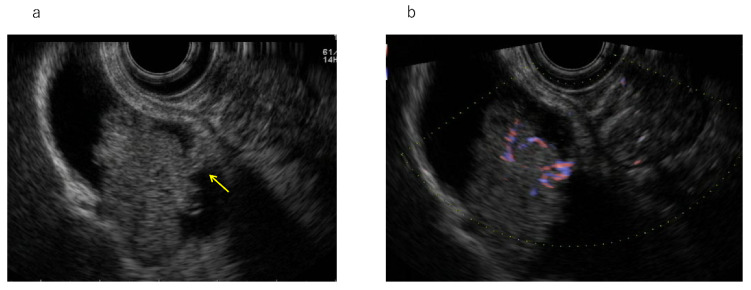
Gallbladder adenoma. (**a**) Endoscopic ultrasonography (EUS): Pedunculated elevated lesion (arrow). (**b**) EUS (Doppler mode): Linear blood flow in the peduncle.

**Figure 3 diagnostics-11-01586-f003:**
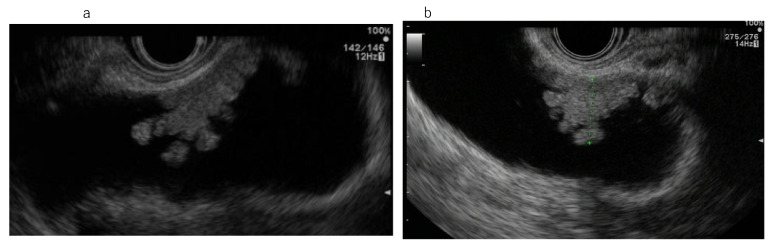
Intracystic papillary neoplasm (ICPN; adenocarcinoma: m). (**a**,**b**) Endoscopic ultrasonography (EUS): Broad-based elevated lesion.

**Figure 4 diagnostics-11-01586-f004:**
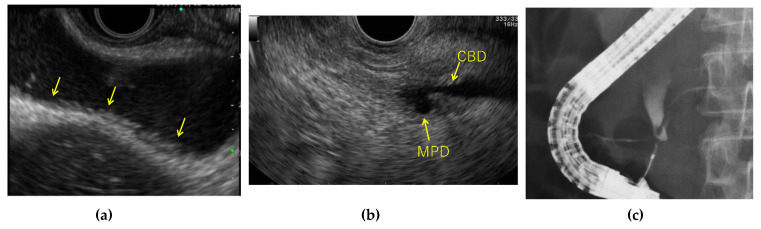
Pancreaticobiliary maljunction with gallbladder wall thickening. (**a**) Endoscopic ultrasonography (EUS): Diffuse low papillary-shaped elevated lesion in the gallbladder (arrow). (**b**) Endoscopic ultrasonography: Pancreaticobiliary duct junction in the pancreatic parenchyma. CBD; common hepatic duct, MPD; main pancreatic duct (**c**) ERP: Findings of the pancreatic duct junction-type maljunction.

**Figure 5 diagnostics-11-01586-f005:**
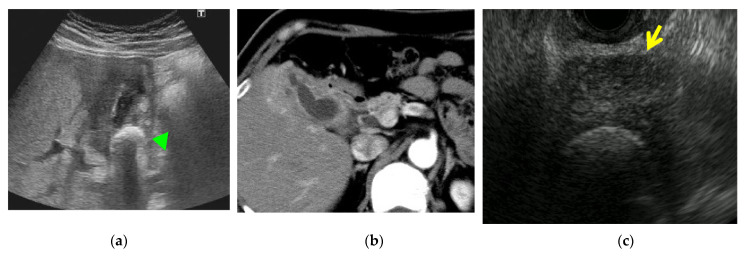
Xanthogranulomatous cholecystitis. (**a**) Abdominal ultrasonography: Wall thickening around the entire gallbladder and gall stone in the neck (arrow head). (**b**) Contrast-enhanced computed tomography: Marked wall thickening and poorly marginated border with liver parenchyma. (**c**) Endoscopic ultrasonography (EUS): Marked wall thickening (arrow).

**Figure 6 diagnostics-11-01586-f006:**
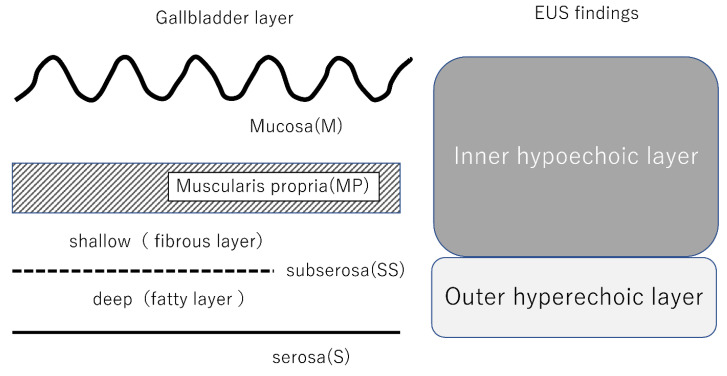
Comparison of gallbladder layer and EUS findings.

**Figure 7 diagnostics-11-01586-f007:**
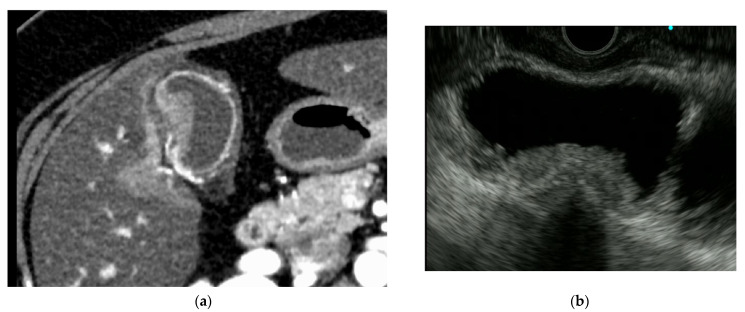
T3 gallbladder cancer. (**a**) Contrast-enhanced computed tomography: enhanced gallbladder mass in the body. (**b**) Endoscopic ultrasonography: hypoechoic tumor in the gallbladder with no hyperechoic outer layer.

**Figure 8 diagnostics-11-01586-f008:**
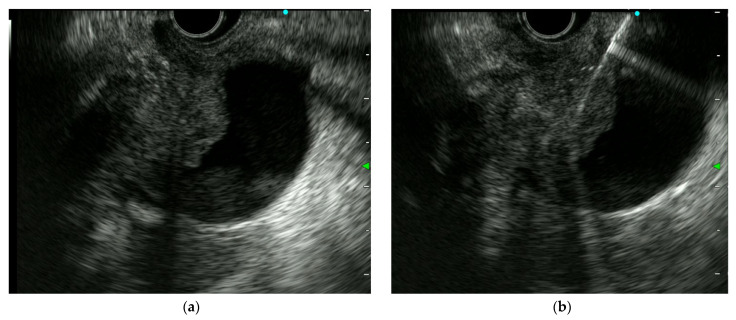
EUS-FNA of the gallbladder. (**a**,**b**) A case of gallbladder cancer: a thickened part of the gallbladder wall punctured while avoiding the gallbladder lumen.

**Table 1 diagnostics-11-01586-t001:** Classification of gallbladder lesion.

		High-to-Intermediate Frequency	Low Frequency
Neoplastic		Adenocarcinoma Adenosquamous carcinoma	Adenoma Carcinosarcoma Metastatic tumor Neuroendocrine tumor Malignant lymphoma
Non-Neoplastic	Elevated lesion	Cholesterol polyp	Hyperplastic polyp Metaplastic polyp Inflammatory polyp Fibrous polyp
	Flat or thickened wall lesion	Cholecystitis Adenomyomatosis Hyperplastic change (associated with the pancreas bile duct maljunction)	

**Table 2 diagnostics-11-01586-t002:** Studies comparing the role of EUS-FNA for the gallbladder mass.

Author	Year	No.	Final Diagnosis	Sensitivity	Specificity	Accuracy	Complication Rate
Jacobson et al. [30]	2003	6	Malignant 5 Benign 1(XGC)	83.3%	100%	83.3%	0%
Varadarajulu et al. [31]	2005	6	Malignant 5 Benign 1(cholecystitis)	100%	100%	100%	0%
Meara et al. [32]	2007	7	Malignant 7	80%	100%	85.71%	0%
Hijioka et al. [23]	2010	15	Malignant 10 Benign 5(XGC)	90%	100%	93.3%	0%
Hijioka et al. [34]	2011	83	Malignant 83	96%	100%	98%	0%
Singla et al. [29]	2018	101	Malignant: 98 Benign 1	90.8%	100%	90.9%	0%

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
