# Peer review of "The Role of EUS and EUS-FNA in Differentiating Benign and Malignant Gallbladder Lesions"

_diagnostics, 2021, doi:10.3390/diagnostics11091586_

Round 1
Reviewer 1 Report
I commend the Authors for this comprehensive review. The manuscript is well-written and accompanied by very nice figures. Technical details are clearly reported making the paper very interesting for expert readers.
I have only a few comments for the Authors.
- EUS-guided tissue acquisition is moving from fine-needle aspiration to fine needle biopsy (FNB). However, in your study, EUS-FNB is not mentioned. A deeper comment on the availability of new needles for EUS-FNB and their results should be included. This is particularly relevant to differentiate adenocarcinomas from sarcomas or lymphomas, and in the view of future genetic analysis on EUS specimens. Doing so, cite the following studies:
- PMID: 34116031
- PMID: 29977995
- PMID: 33390343
- PMID: 31070037
- PMID: 30484917
- Please, clarify the reason why “When taking a sample from the gallbladder, it is critical that the needle not pass through the lumen of the gallbladder”. Moreover, the readers could ask themselves why the new through-the-needle biopsy microforceps cannot be used in this setting (the gallbladder could be considered a sort of cyst). Please explain this point. Doing so refers to PMID: 31323232.
Author Response
accompanied by very nice figures. Technical details are clearly reported making the paper very interesting for expert readers.I have only a few comments for the Authors.
- EUS-guided tissue acquisition is moving from fine-needle aspiration to fine needle biopsy (FNB). However, in your study, EUS-FNB is not mentioned. A deeper comment on the availability of new needles for EUS-FNB and their results should be included. This is particularly relevant to differentiate adenocarcinomas from sarcomas or lymphomas, and in the view of future genetic analysis on EUS specimens. Doing so, cite the following studies:
- PMID: 34116031
- PMID: 29977995
- PMID: 33390343
- PMID: 31070037
- PMID: 30484917
Thank you for your valuable comment; I completely agree with it. According to your comment, I have added the following sentences.
“Recently, novel fine-needle biopsy (FNB) devices have been specifically designed to obtain histological specimens. Histological specimens with preserved architecture are easier to be interpreted by a standard pathologist than cytological smears.
Indeed, recent results of EUS-FNB for both pancreatic and non-pancreatic lesions demonstrated that the histological yield and accuracy rate are excellent. Therefore, it is recommended to use EUS-FNB, in particular, for the examination of cases that require IHC, including the metastasis from another primary organs’ malignancy/a malignant lymphoma, and those whose genetic analyses have been described below.”
- Please, clarify the reason why “When taking a sample from the gallbladder, it is critical that the needle not pass through the lumen of the gallbladder”. Moreover, the readers could ask themselves why the new through-the-needle biopsy microforceps cannot be used in this setting (the gallbladder could be considered a sort of cyst). Please explain this point. Doing so refers to PMID: 31323232.
Thank you for your valuable comment. I am extremely sorry for the typographical error. I have changed the word and added the following sentence.
“it is important that the needle does not pass through the lumen of the gallbladder because bile leakage from the needle puncture site may cause peritonitis and peritoneal dissemination in cases of malignancy.”
Besides, I have discussed about the new through-the-needle biopsy microforceps under the subheading “EUS-FNA Indications for Differential Diagnosis of Gallbladder Lesions”, as follows.
“Recently, a through-the-needle microforceps device was developed for EUS-guided tissue sampling of pancreas cystic lesions. These microforceps can be advanced through the lumen of a standard 19-gauge EUS-FNA needle for through-the-needle tissue biopsy (TTNB) of pancreas cystic lesions. EUS-TTNB is a useful tool for the differential diagnosis of pancreatic cystic lesions. It can also be applied to lesions in the gallbladder, although there is no report so far. However, about the complications, one technical review noted complications ranging from 2 to 23%. Moreover, EUS-TTNB is thought to have the risk of gallbladder perforation because the gallbladder itself is cystic, and the gallbladder wall is thin. Therefore, we think that the indication of EUS- TTNB for the gallbladder is very restrictive at present.”
Reviewer 2 Report
1. In the second paragraph sentence two must be corrected. The classification of gallblader lesions as neoplastic or non-neoplastic is not based on their frequency but on their microscopic structure and of their invasive characteristics.
2. Third Paragraph, page 3/13: Cholesterol Polyp are typically small ( < 4 mm, not 10 mm ).
3. Why do you compare the sonographic density of gallbladderpolyps with liver parenchyma? Gallbladderpolyps have typically the same densitiy like the gallbladderwall.
4. Paragraph 4: Sonozoid@ is not available in severeal countries, i. e. Germany. Please add SonoVue@ as ultrasound contrast medium.
5. Paragraph 5, section 3: Sensitivity and specifity of EUS-FNA is high only in sciientific Settings. A comparison with real wolrd data would make sense, due to the much worse data in this area.
6. Paragraph 5.1: EUS-FNA an Dissemination of malignant cells. Please explain more precisey and consider form metstatic outspread after EUS-FNA.
Author Response
- In the second paragraph sentence two must be corrected. The classification of gallblader lesions as neoplastic or non-neoplastic is not based on their frequency but on their microscopic structure and of their invasive characteristics.
Thank you for your valuable comment. I am extremely sorry for the typographical error.
- Third Paragraph, page 3/13: Cholesterol Polyp are typically small ( < 4 mm, not 10 mm ).
Thank you for your valuable comment. I have changed 10 mm to ≦4 mm.
- Why do you compare the sonographic density of gallbladderpolyps with liver parenchyma? Gallbladderpolyps have typically the same densitiy like the gallbladderwall.
Thank you for your valuable comment. As for liver parenchyma, I referred to the ref. Journal of Ultrasound pages 131–142 (2021). I have added gall bladder wall considering your valuable comment.
- Paragraph 4: Sonozoid@ is not available in severeal countries, i. e. Germany. Please add SonoVue@ as ultrasound contrast medium.
Thank you for your valuable comment. I have added SonoVue@ as the ultrasound contrast medium.
- Paragraph 5, section 3: Sensitivity and specifity of EUS-FNA is high only in sciientific Settings. A comparison with real wolrd data would make sense, due to the much worse data in this area.
Thank you for your valuable comment. I agree with your comment. However, we have included the worse data as well. Accordingly, I have added the following sentence "However, we have to pay attention to the high sensitivity and specificity rates because of the possibility of a selective outcome reporting bias.", as you had pointed out.
- Paragraph 5.1: EUS-FNA an Dissemination of malignant cells. Please explain more precisey and consider form metstatic outspread after EUS-FNA.
Thank you for your valuable comment. I agree with your comment. According to your comment, I have added the following sentences.
“Recently, cases of dissemination needle-tract seeding caused by EUS-FNA have been reported, most of which are needle-tract seeding being intragastric wall metastases, and peritoneal dissemination is rare. Though the frequency of dissemination is extremely low, peritoneal dissemination via EUS-FNA is an issue because it impairs patient survival.”